# Enhanced Corrosion Resistance and Biological Properties of Ultrafine-Grained Ti15Zr5Cu Alloy

**Hai Wang** [1,†], **Wenwei Gao** [1,2,†], **Xiyue Zhang** [1,2], **Yi Li** [1,2], **Shuyuan Zhang** [1], **Ling Ren** [1,3,*] and **Ke Yang** [1]

1   Shi-Changxu Innovation Center for Advanced Materials, Institute of Metal Research, Chinese Academy of Sciences, Shenyang 110016, China; hwang13s@imr.ac.cn (H.W.); wwgao19s@imr.ac.cn (W.G.); xyzhang19s@imr.ac.cn (X.Z.); yli20b@imr.ac.cn (Y.L.); syzhang@imr.ac.cn (S.Z.); kyang@imr.ac.cn (K.Y.)
2   School of Materials Science and Engineering, University of Science and Technology of China, Shenyang 110016, China
3   Binzhou Institute of Technology, Shandong Key Laboratory of Advanced Aluminium Materials and Technology, Weiqiao-UCAS Science and Technology Park, Binzhou 256606, China
*   Correspondence: lren@imr.ac.cn
†   These authors contributed equally to this work.

**Abstract:** Titanium alloys are widely used in the biomedical field. To ensure their strength meets requirements in clinics, medical titanium alloys are generally alloyed with toxic Al and/or V elements, hence ensuring their long-term biological safety after implantation is a challenge. In our previous research, we developed an ultrafine-grained Ti15Zr5Cu alloy without toxic elements while its mechanical properties were at the same level with the most widely used Ti6Al4V alloy. In order to promote the clinical application of the ultrafine-grained Ti15Zr5Cu alloy, herein we have systematically studied the hot deformation behaviors of the material as well as evaluated its corrosion resistance and biological properties. Results showed that when the as-quenched Ti15Zr5Cu alloy deformed at $0.05 \leq \dot{\varepsilon} \leq 1$, $730\,^{\circ}\text{C} \leq \text{T} \leq 750\,^{\circ}\text{C}$, it not only possessed good workability but also can be converted into an equiaxed ultrafine-grained microstructure. Moreover, the material also exhibited better corrosion resistance, antibacterial properties and biocompatibility than the Ti15Zr alloy and the commercial pure Ti. The results of the present study help lay a foundation for the development of a new generation of medical titanium alloys.

**Keywords:** ultrafine-grained; hot workability; corrosion resistance; antibacterial properties; cytotoxicity

## 1. Introduction

Titanium alloys have been widely used in the biomedical field due to their high specific strength, good corrosion resistance and excellent biocompatibility [1]. Nowadays, most of the titanium alloys used in orthopedics or dentistry, such as the Ti6Al4V alloy and the Ti6Al7Nb alloy, are alloyed with toxic Al and/or V elements [2,3], hence their long-term biosafety after implantation raises concerns. For those titanium alloys without toxic elements, such as commercial pure Ti or the Ti15Zr alloy [4,5], their strength is too low to meet clinical demands. In our previous study, we developed an ultrafine-grained Ti15Zr5Cu alloy with comparable mechanical properties to the widely used Ti6Al4V alloy [6] which is anticipated to address the problem with current medical titanium alloys.

The fabrication of the aforementioned ultrafine-grained Ti15Zr5Cu alloy is based on a eutectoid element alloying-quenching-hot deformation (EQD) strategy [6,7]. By being massively alloyed with the eutectoid element Cu then quenching from the single $\beta$ phase region at elevated temperatures, Cu would supersaturate and dissolve in the matrix. This would cause severe lattice distortion to be stored in the newly formed $\alpha'$ martensite microstructure. Through hot deformation with the $\alpha'$ microstructure, it would then decompose into $\alpha$ and Ti$_2$Cu phases. In this process, the nucleation rate of dynamic

recrystallization in the $\alpha$ phase would prominently increase due to high strain energy being stored in the initial $\alpha'$ microstructure. Meanwhile, $Ti_2Cu$ particles would dissolve from the matrix and would exert Zenner pinning force on the grain boundaries to inhibit $\alpha$ grains becoming coarsened at elevated temperatures. It should be noted that in the EQD fabrication process, the hot deformation temperature and the deformation strain rate play a key role in acquiring an ultrafine-grained microstructure [6,7]. Inappropriate deformation parameters might cause the formation of coarse grains, hence decreasing the mechanical properties or resulting in the formation of holes and microcracks which would thus limit the subsequent machining of the material [8]. Therefore, it is imperative to systematically study the hot deformation behavior of the as-quenched Ti15Zr5Cu alloy and to determine the optimal hot deformation window for its EQD fabrication process.

Ultrafine-grained Ti15Zr5Cu alloy possesses satisfactory mechanical properties; however, its corrosion resistance and biological properties deserve to be further evaluated. Firstly, a large number of studies showed that grain refinement would increase the thickness of passive film on the surface of the titanium alloys, which would improve uniform corrosion resistance [9–11]. Some research results also hinted that a $Ti_2Cu$ phase precipitated along grain boundaries would form numerous tiny galvanic cells [12,13], which gave rise to galvanic corrosion and sharply reduced the pitting corrosion resistance of the material. Secondly, studies showed that ultrafine-grained microstructures bring about high surface energy to titanium alloys, which promotes the adhesion of osteoblasts and accelerated bone integration after implantation [14,15]. However, high surface energy of a material would also promote bacterial adhesion and increase the risk of infection after implantation surgeries [16]. Last but not least, studies also indicated that for Cu-bearing medical metals the continuous dissolved Cu ions from the material surface endowed it with multiple biological functions, such as antibacterial properties, promoting osteogenesis, angiogenesis and endothelialization [17,18]. Nevertheless, excessive Cu ions might also lead to cytotoxicity [19]. In sum, it is of great importance to study the corrosion resistance and biological properties of the ultrafine-grained Ti15Zr5Cu alloy.

In this research, we will first systematically investigate the hot compression behavior of the as-quenched Ti15Zr5Cu alloy under different temperatures and strain rates, then calculate the constitutive equation and processing maps and characterize its microstructure after hot compression. Hence, the optimum hot deformation parameters for the EQD fabrication process can be determined. On this basis, the ultrafine-grained Ti15Zr5Cu alloy can be prepared through conventional hot rolling technology. Finally, the corrosion resistance, antibacterial properties and cytotoxicity of the alloy will be evaluated. The results of this study will be of great significance in promoting the application of the ultrafine-grained Ti15Zr5Cu alloy in clinics.

## 2. Materials and Experimental Procedures

### 2.1. Processing Window for the Ti15Zr5Cu Alloy

The experimental Ti15Zr5Cu alloy was melted using a 15 kg vacuum arc remelting furnace. The ingot was then held at 1200 °C for 2 h and forged into 60 × 90 × 400 mm billet. The end forging temperature was higher than 1100 °C. After hot forging, it was immediately quenched to room temperature in a 30 wt.% NaCl water solution. Standard cylindrical samples with a diameter of 8 mm and a height of 12 mm were machined from the billet and a hot compression test was then conducted on a Gleeble-3800 thermomechanical simulator machine (DSI, St. Paul, Minnesota, USA) at 710 °C, 750 °C, 790 °C and 830 °C with strain rates of $0.01 \text{ s}^{-1}$, $0.1 \text{ s}^{-1}$, $1 \text{ s}^{-1}$, $10 \text{ s}^{-1}$, respectively. The height reduction in the compression test is 70%, corresponding to the true strain of 1.2. According to the flow stress–strain curves, we obtained peak flow stress and stress at different strains from which we could calculate the constitutive equation and processing maps. A TESCAN MIRA3 scanning electron microscope (TESCAN, Brno, Czech Republic) was used to characterize the microstructure in the central part of each as-deformed specimen, and in combination with the processing map the hot processing window for the Ti15Zr5Cu alloy can be determined.

## 2.2. Corrosion Resistance and Biological Properties

Based on the window condition, the ultrafine-grained Ti15Zr5Cu alloy was fabricated through conventional hot rolling technology. The microstructure of the as prepared material was characterized by a Talos F200X transmission electron microscope (Thermo Scientific, Waltham, MS, USA). Standard $\varphi$3 mm cylindrical tensile samples were machined along the rolling direction (RD) and transverse direction (TD). The tensile properties were given by 3 parallel samples tested by the electromechanical universal testing machine. A Gamry reference 600 workstation was used to test the electrochemical performance of the material in phosphate buffer solution (PBS) at 37 °C, pH = 7.2. The potentiodynamic polarization (PD) curves and electrochemical impedance spectroscopy (EIS) of the material were analyzed. Escherichia coli (*E. coli*, *CICC 10899*) with a concentration of $4.6 \times 10^5$ cfu/mL and Staphylococcus aureus (*S. aureus*, *ATCC 6538*) with a concentration of $6.4 \times 10^5$ cfu/mL were used to evaluate the antibacterial properties of the material. An antibacterial test was executed according to the GB/T 31402-2015 standard. A CCK-8 Kit and mouse embryonic osteoblasts MC3T3-E1 were used to evaluate the effect of the material on cell proliferation. Optical density (OD) value was measured by a microplate reader which can reflect the cytotoxicity of the material. Rhodamine B marked phalloidin and DAPI were used to dye the cytoskeleton on the surface of the materials, and the morphology of cells of different materials was observed by laser confocal microscope.

## 3. Results and Discussions

### 3.1. Flow Behavior

Flow stress–strain curves of the as-quenched Ti15Zr5Cu alloy under different deformation conditions are shown in Figure 1. It can be seen that flow stresses always decreased with the increase in deformation temperature, or with the decrease in deformation strain rate. According to the shapes of the stress–strain curves, the material experienced three stages during the hot deformation. In the first stage ($0 < \varepsilon \leq 0.08$), material exhibited a strain-hardening behavior due to the increase in dislocation density in the material, and the intersection of gliding dislocations led to a continuous increase in flow stress [20]. In the second stage ($0.08 < \varepsilon \leq 0.6$), dynamic recovery (DRV) and dynamic recrystallization (DRX) occurred and flow stress thus decreased with the increase in strain. In the third stage ($\varepsilon > 0.6$), flow stress almost maintained a constant value which is due to strain hardening and DRV/DRX softening reaching a balanced state [21].

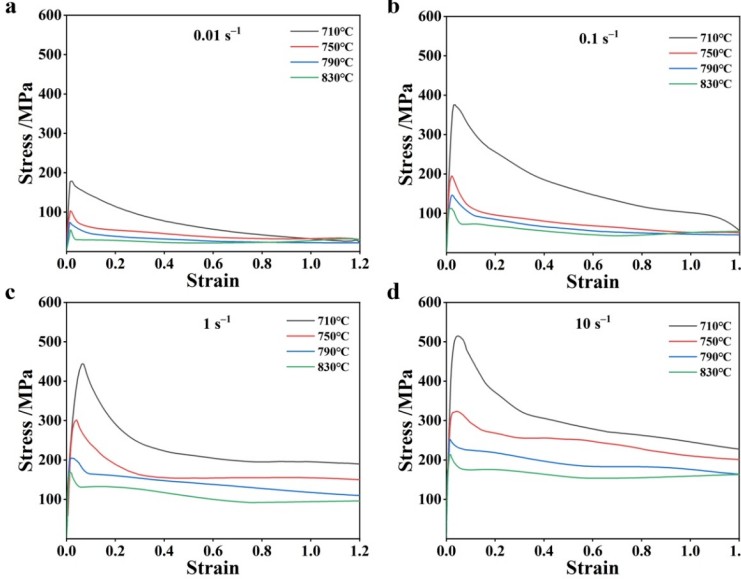

**Figure 1.** Flow stress–strain curves of Ti15Zr5Cu alloy under a constant strain rate and different temperatures. (**a**) 0.01 s$^{-1}$; (**b**) 0.1 s$^{-1}$; (**c**) 1 s$^{-1}$; (**d**) 10 s$^{-1}$.

### 3.2. Constitutive Equation

According to the flow curves (Figure 1), peak flow stress $\sigma$ (MPa) under different deformation conditions can be obtained, as shown in the Table 1. At different peak flow stress levels, the constitutive equation can be expressed as follows [22]:

$$Z = \dot{\varepsilon}\exp(Q/RT) = A\sigma^{n'} \quad (\alpha\sigma < 0.8) \tag{1}$$

$$= B\exp(\beta\sigma) \quad (\alpha\sigma \geq 1.2) \tag{2}$$

$$= C[\sinh(\alpha\sigma)]^{n} \quad (\text{for all } \sigma) \tag{3}$$

where $Z$ (s$^{-1}$) represents the Zener–Hollomon parameter, $\dot{\varepsilon}$ (s$^{-1}$) represents strain rate, $Q$ (kJ·mol$^{-1}$) is the hot deformation activation energy, $T$ (K) is the absolute temperature, R (8.314 J·mol$^{-1}$·K$^{-1}$) is the universal gas constant, A, B and C (s$^{-1}$) are material constants, n and n$'$ are stress exponents, and $\alpha = \beta/n'$ (MPa$^{-1}$) and $\beta$ (MPa$^{-1}$) are stress coefficients. In order to determine the parameters in the constitutive equations, the natural logarithm was taken on both sides of Equations (1)–(3), and we can thus obtain the following equations:

$$ln\dot{\varepsilon} = lnA + n'ln\sigma - Q/RT \tag{4}$$

$$ln\dot{\varepsilon} = lnB + \beta\sigma - Q/RT \tag{5}$$

$$ln\dot{\varepsilon} = lnC + nln[\sinh(\alpha\sigma)] - Q/RT \tag{6}$$

with a fixed deformation temperature. Taking the partial differentiation of Equations (4)–(6), the following relationships can be derived:

$$n' = \partial ln\dot{\varepsilon}/\partial ln\sigma\big|_{T} \tag{7}$$

$$\beta = \partial ln\dot{\varepsilon}/\partial\sigma\big|_{T} \tag{8}$$

$$n = \partial ln\dot{\varepsilon}/\partial ln[\sinh(\alpha\sigma)]\big|_{T} \tag{9}$$

**Table 1.** Peak flow stress of as-quenched Ti15Zr5Cu alloy under different deformation conditions.

| Temperature \ Strain Rate | 0.01 s$^{-1}$ | 0.1 s$^{-1}$ | 1 s$^{-1}$ | 10 s$^{-1}$ |
|---|---|---|---|---|
| 710 °C | 178.62 | 375.22 | 446.80 | 521.59 |
| 750 °C | 103.61 | 194.51 | 301.55 | 325.83 |
| 790 °C | 73.878 | 144.56 | 205.16 | 252.04 |
| 830 °C | 53.50 | 111.42 | 168.52 | 213.25 |

According to Equations (7) and (8), $ln\dot{\varepsilon} - ln\sigma$ and $ln\dot{\varepsilon} - \sigma$ scatter diagrams were plotted and linearly fitted (Figure 2a,b). The acquired slopes were taken as the value of n$'$ = 5.32 and $\beta$ = 0.03222 MPa$^{-1}$, respectively. According to Equation (9), n = 3.76 was then determined by the slopes of $ln\dot{\varepsilon} - ln[\sinh(\alpha\sigma)]$ curves (Figure 2c). Based on Equation (6), hot deformation activation energy at a constant strain rate can be expressed by:

$$Q = Rn\partial ln[\sinh(\alpha\sigma)]/\partial(1/T)\big|_{\dot{\varepsilon}} \tag{10}$$

The $Q$ value can be calculated by the slope of $ln[\sinh(\alpha\sigma)] - /T$ curves (Figure 2d). Finally, according to Equation (6), the C value can be calculated by the intercept of $ln\dot{\varepsilon} + Q/RT - nln[\sinh(\alpha\sigma)]$ curves. Hence, the constitutive equation of quenched Ti15Zr5Cu alloy can be expressed as:

$$Z = \dot{\varepsilon}\exp(491500/RT) = 1.93 \times 10^{23}[\sinh(0.0062\sigma)]^{3.76} \tag{11}$$

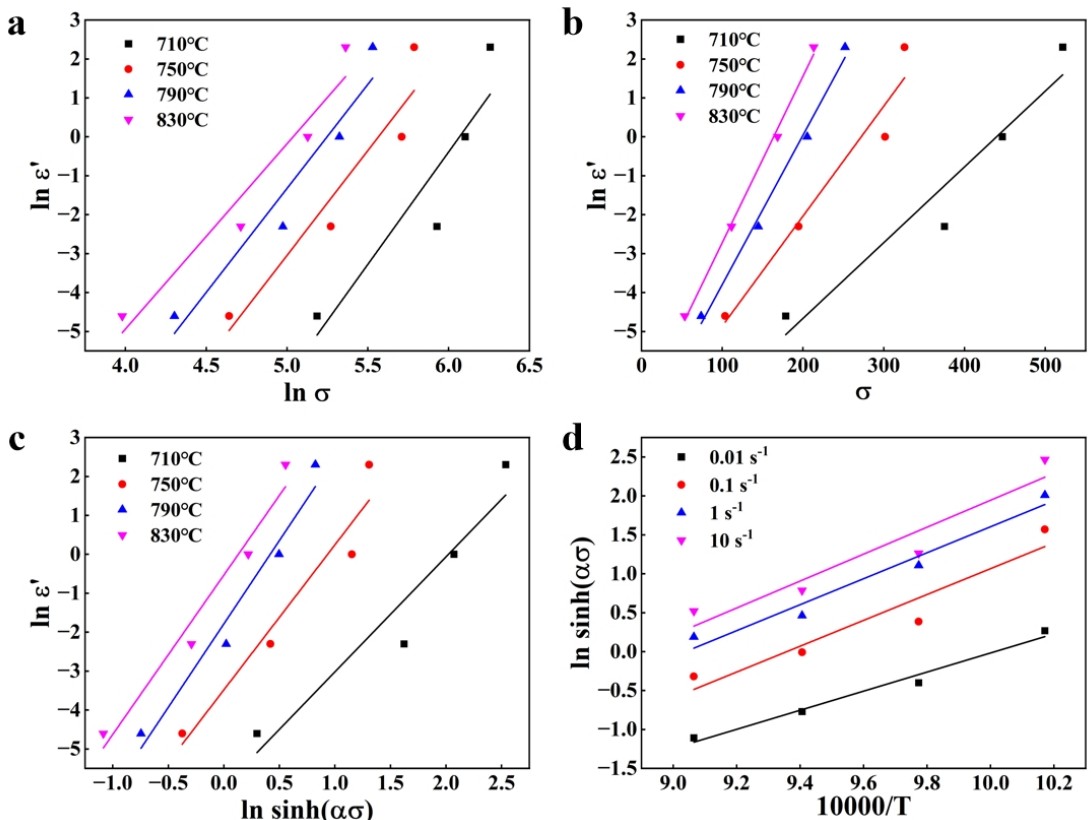

**Figure 2.** Linear fitting data for determining parameters in the constitutive equation. (**a**) $ln\dot{\varepsilon} - ln\sigma$ curves; (**b**) $ln\dot{\varepsilon} - \sigma$ curves; (**c**) $ln\dot{\varepsilon} - ln[\sinh(\alpha\sigma)]$ curves; (**d**) $ln[\sinh(\alpha\sigma)] - 1/T$ curves.

According to the constitutive equation, flow stress of the material at certain temperatures and strain rates during deformation could be forecasted so that it is possible to provide theoretical guidance for the practical hot working of the material. From Equation (11), the $Q$ value for the as-quenched Ti15Zr5Cu alloy reaches 491.5 kJ·mol$^{-1}$, which is much higher than that of the self-diffusion activation energy of $\alpha$-Ti of 169~242 kJ·mol$^{-1}$ [23,24]. This can be attributed to two reasons. On the one hand, the element Zr in the material reduced the self-diffusion coefficient of the matrix due to its high binding energy, which slowed down the DRV rate of the microstructure and made it more difficult to deform. On the other hand, the precipitated Ti$_2$Cu particles during the deformation would pin grain boundaries and thus lead to higher deformation activation energy.

### 3.3. Microstructure Evolution during Hot Deformations

The microstructure of the hot deformed Ti15Zr5Cu alloy was characterized by SEM, as shown in Figure 3. When deformed at 750 °C and with a low deformation strain rate, the DRX process was highly promoted and the initial martensite microstructure was completely converted into an equiaxed ultrafine-grained microstructure. When deformed under the other conditions, the initial martensite microstructure was recovered and lath width increased obviously. Therefore, a coarse lamellar microstructure formed after the hot deformation.

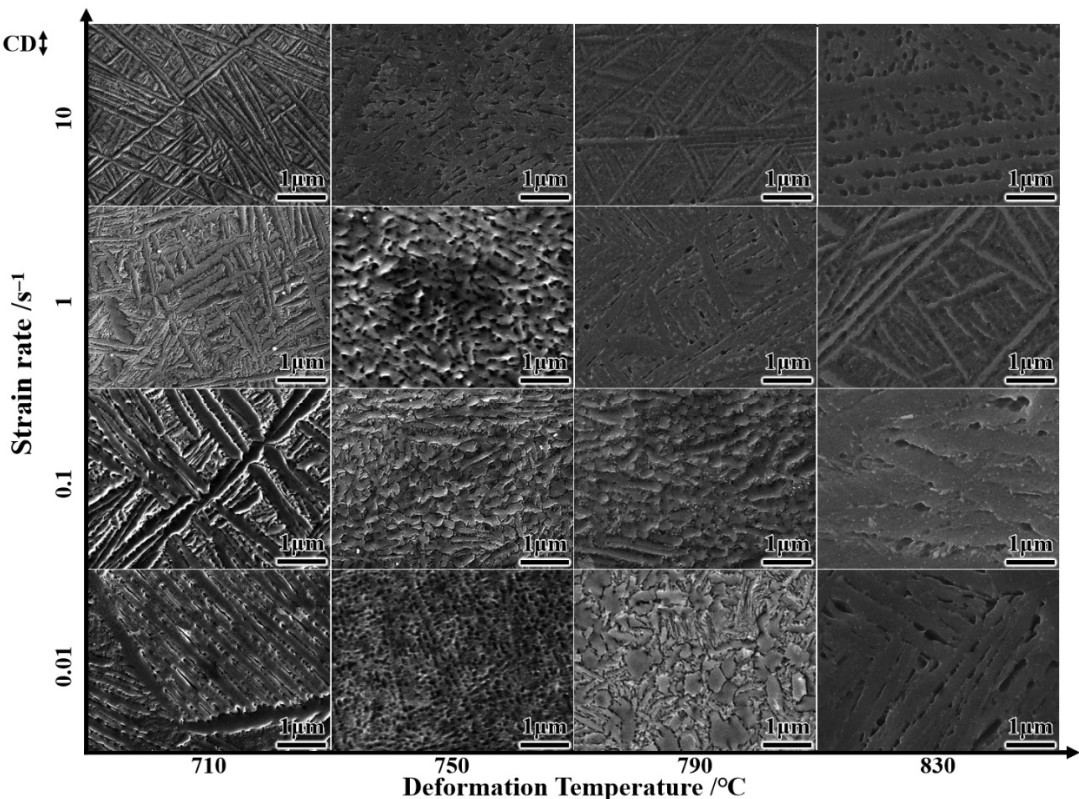

**Figure 3.** SEM observations of deformed Ti15Zr5Cu alloy under different conditions. CD represents compression direction.

### 3.4. Processing Map

According to Figure 1, the flow stress of the Ti15Zr5Cu alloy at different strains during hot deformations can be determined as shown in Table 2, from which we can calculate the strain rate sensitivity parameter *m* based on the following equation [25]:

$$m = \frac{\partial ln\sigma}{\partial ln\dot{\varepsilon}}|_{\varepsilon,\mathrm{T}} \tag{12}$$

Thus, the energy dissipation rate $\eta$ and the flow instability parameter $\xi$ were calculated based on the following equations [25]:

$$\eta = \frac{2m}{m+1} \tag{13}$$

$$\xi = \frac{\partial ln\left(\frac{m}{m+1}\right)}{\partial ln\dot{\varepsilon}} + m \tag{14}$$

where $\eta$ reflects the ratio of energy dissipated by microstructure evolution to total input energy during the hot deformation; a higher $\eta$ value means better hot workability. $\xi$ reflects flow instability tendency during the hot deformation. When $\xi < 0$, the material is prone to present adiabatic shear bands or microcracks. Through superposing the $\eta$ contour map with the instability regime map ($\xi < 0$, marked in pink), we can then obtain the processing map for the Ti15Zr5Cu alloy (Figure 4). Based on the SEM results and processing maps, it can be seen that when the material deformed at conditions of $0.05 \leq \dot{\varepsilon} \leq 1$ and $730\,°C \leq T \leq 750\,°C$ (orange area), it would not only obtain an equiaxed ultrafine-grained microstructure, but also develop excellent hot workability.

**Table 2.** Flow stress (MPa) of the Ti15Zr5Cu alloy at different strain rates under various deformation conditions.

| Strain | Temperature | 0.01 s$^{-1}$ | 0.1 s$^{-1}$ | 1 s$^{-1}$ | 10 s$^{-1}$ |
|---|---|---|---|---|---|
| 0.2 | 710 °C | 114 | 256 | 291 | 372 |
| | 750 °C | 54 | 96 | 189 | 268 |
| | 790 °C | 38 | 84 | 160 | 218 |
| | 830 °C | 28 | 67 | 131 | 176 |
| 0.4 | 710 °C | 77 | 185 | 223 | 306 |
| | 750 °C | 46 | 80 | 155 | 256 |
| | 790 °C | 32 | 66 | 147 | 197 |
| | 830 °C | 23 | 55 | 117 | 164 |
| 0.6 | 710 °C | 56 | 147 | 223 | 279 |
| | 750 °C | 36 | 68 | 154 | 247 |
| | 790 °C | 26 | 55 | 138 | 184 |
| | 830 °C | 22 | 46 | 100 | 154 |
| 0.8 | 710 °C | 42 | 117 | 195 | 263 |
| | 750 °C | 33 | 59 | 155 | 228 |
| | 790 °C | 24 | 50 | 128 | 183 |
| | 830 °C | 23 | 45 | 92 | 155 |
| 1.0 | 710 °C | 32 | 101 | 195 | 246 |
| | 750 °C | 32 | 52 | 155 | 211 |
| | 790 °C | 22 | 47 | 118 | 176 |
| | 830 °C | 28 | 51.3 | 94 | 159 |

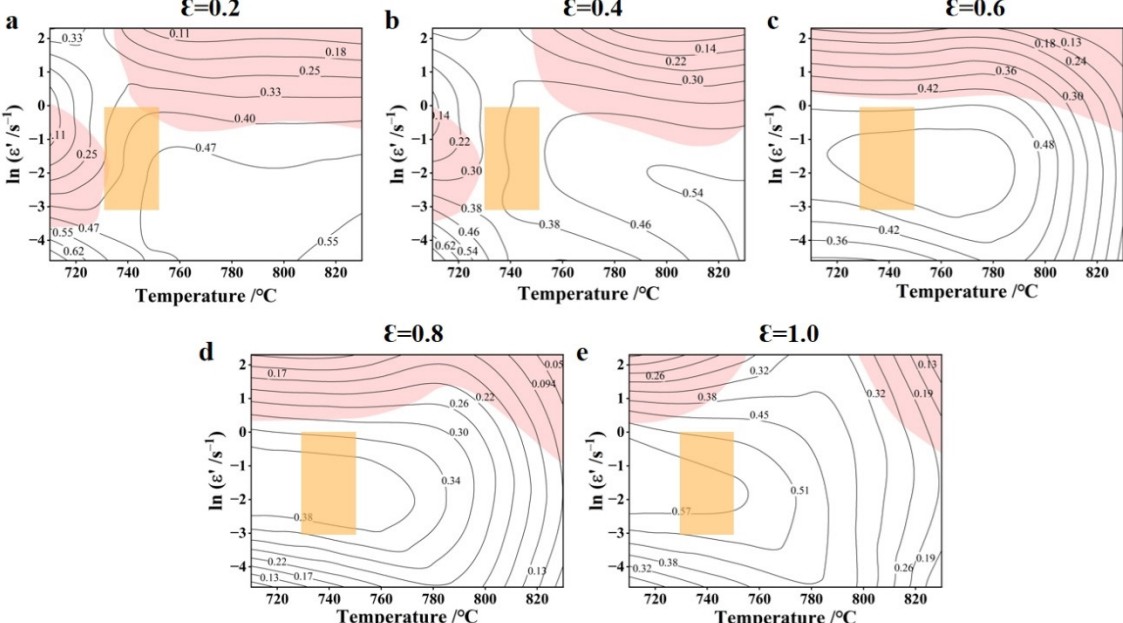

**Figure 4.** Processing map of the Ti15Zr5Cu alloy at different strain rates. (**a**) $\varepsilon$ = 0.2; (**b**) $\varepsilon$ = 0.4; (**c**) $\varepsilon$ = 0.6; (**d**) $\varepsilon$ = 0.8; (**e**) $\varepsilon$ = 1.0. The optimal deformation window is marked in orange.

### 3.5. Microstructure and Mechanical Property of the As-Rolled Ti15Zr5Cu Alloy

The forged billet was hot rolled into a sheet under the aforementioned optimal hot deformation condition. The microstructure of the material was then characterized by the Transmission electron microscopy (TEM). As shown in Figure 5a, the grain size of the as-deformed alloy is in a range of 100~300 nm. According to the tensile curves in Figure 5b, the tensile property of the ultrafine-grained Ti15Zr5Cu alloy is at the same level as the biomedical Ti6Al4V alloy, and the dominant strengthening mechanism of the alloy

should be grain refinement strengthening. Compared to the biomedical Ti6Al4V alloy, the ultrafine-grained Ti15Zr5Cu alloy is chemically safe without toxic Al and V elements, thus it is beneficial for improving long-term biological safety after implantation surgery.

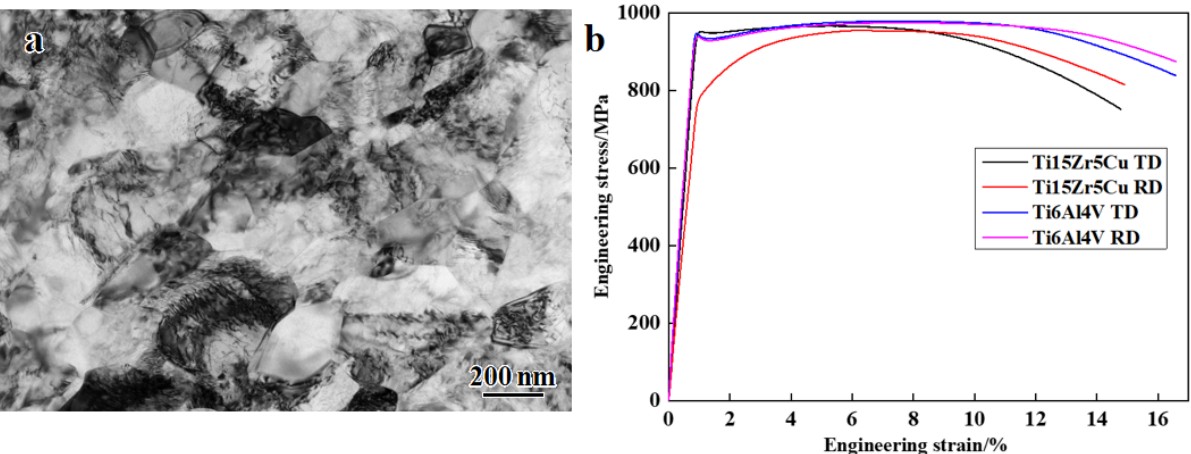

**Figure 5.** Microstructure and tensile property of the as-deformed ultrafine-grained Ti15Zr5Cu alloy. (**a**) TEM observation; (**b**) room temperature tensile curves.

### 3.6. Corrosion Resistance

Electrochemical tests were conducted on the ultrafine-grained Ti15Zr5Cu alloy, the commercial pure Ti and the Ti15Zr alloy. Potential polarization curves, as shown in Figure 6a, indicated that the ultrafine-grained Ti15Zr5Cu alloy exhibited a higher corrosion resistance with higher corrosion potential $E_{corr}$ and lower corrosion current density $I_{corr}$. Bode diagrams (Figure 6b,c) based on EIS analysis indicated that, for the three measured materials, the phase angle at a medium frequency (1~100 Hz) was close to $-80°$ and the slope of $\log|z|\sim\log f$ curves for 0.01~100 Hz was close to $-1$, reflecting that they possessed stable passive films. For the ultrafine-grained Ti15Zr5Cu alloy, the frequency range corresponding to the maximum phase angle was wider than that of the commercial pure Ti and the Ti15Zr alloy, and its $\log|z|$ value at low and medium frequency (0.01~100 Hz) was always higher than the other two materials, indicating it possessed a denser passive film. Nyquist curves of the ultrafine-grained Ti15Zr5Cu alloy showed a smaller curvature to the commercial pure Ti and the Ti15Zr alloy (Figure 6d), which hinted that it had a larger polarization resistance and thus the electrode reaction in the passive film was proceeded with more difficulty. The impedance spectrum was fitted using the equivalent circuit in Figure 6d, and the fitting results are shown in Table 3. Solution resistance $R_s$ for all three materials were at a similar level, while the passive film of the ultrafine-grained Ti15Zr5Cu alloy possessed a larger resistance $R_{ct}$ and a lower capacitance Q, so the passive film could effectively protect the material from corrosion. Based on the above results, it can be seen that although Cu alloying can give rise to $Ti_2Cu$ phase precipitation along grain boundaries, their sizes were too small to reduce the pitting corrosion resistance of the material [12,13]. Furthermore, the ultrafine-grained microstructure can significantly improve the compactness of the passive film, endowing the Ti15Zr5Cu alloy with better corrosion resistance [9–11].

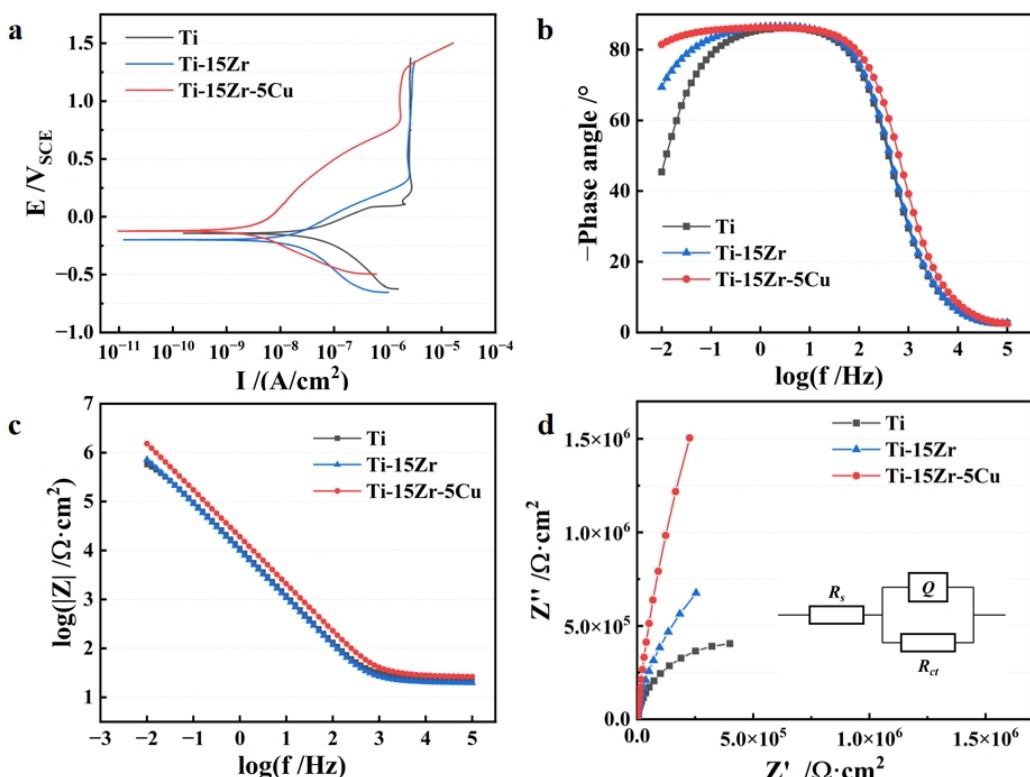

**Figure 6.** Electrochemical tests results in PBS at 37 °C and pH = 7.2. (**a**) PD curves; (**b,c**) Bode diagram; (**d**) Nyquist curves and equivalent circuit.

**Table 3.** PD curves analysis and equivalent circuit fitting results.

|  | $I_{corr}$/nA·cm$^{-2}$ | $E_{corr}$/V | $R_s$/Ω·cm$^{-2}$ | Q/μF·cm$^{-2}$ | $R_{ct}$/Ω·cm$^{-2}$ |
|---|---|---|---|---|---|
| Ti | 61.06 | −0.144 | 24.48 | 15.1 | $8.6 \times 10^5$ |
| Ti-15Zr | 27.55 | −0.198 | 20.75 | 17.6 | $2.3 \times 10^6$ |
| Ti-15Zr-5Cu | 3.13 | −0.121 | 27.88 | 9.1 | $1.8 \times 10^7$ |

### 3.7. Antibacterial Property

Antibacterial properties of the as-deformed Ti15Zr5Cu alloy were tested and the results are shown in Figure 7. After co-culture with *E. coli* and *S. aureus* for 24 h, almost no colony was observed on the slab of the ultrafine-grained Ti15Zr5Cu alloy, while plentiful colonies were found on the slabs of the control group Ti and Ti15Zr alloy. Quantitative analysis showed that the antibacterial rate of the ultrafine-grained Ti15Zr5Cu alloy against *E. coli* and *S. aureus* was above 99%. Its excellent antibacterial performance is mainly due to the continuous minor release of $Cu^{2+}$ from the surface of the material which has a charge interaction with the bacteria and changes the permeability of the cell membrane. $Cu^{2+}$ also influences the sulfhydryl group in the respiratory enzyme around the cell membrane, resulting in obstructed cellular respiration which thus affects normal physiological function and proliferation of the bacteria; the bacteria are killed because of membrane leakage and cytoplasm outflow [26–28]. In sum, the antibacterial property of the ultrafine-grained Ti15Zr5Cu alloy will effectively reduce the risk of infection, which is of great significance for improving success rates for implantation surgery.

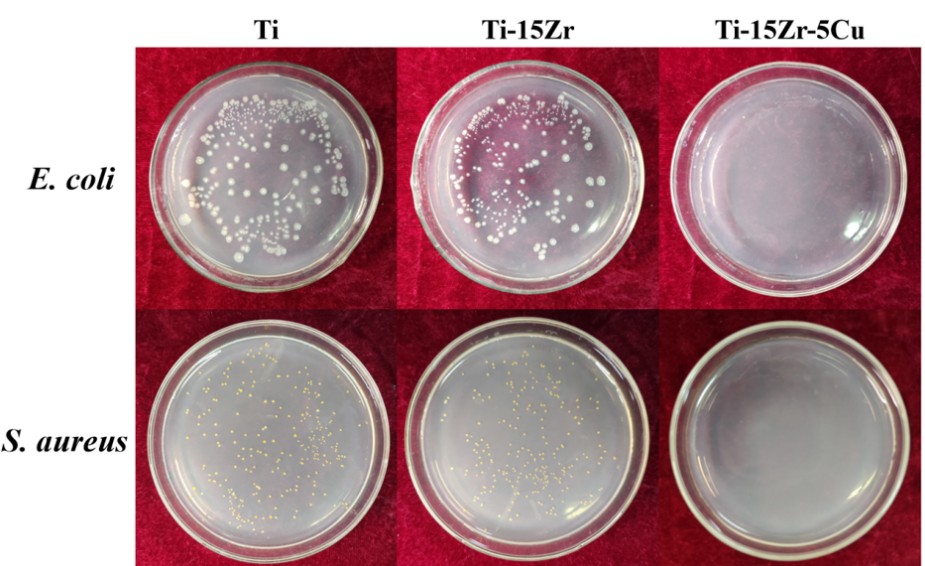

**Figure 7.** Antibacterial test results of the commercial pure Ti, the Ti15Zr alloy and the ultrafine-grained Ti15Zr5Cu alloy co-cultured with *E. coli* and *S. aureus* for 24 h.

*3.8. Cytotoxicity Test*

The ultrafine-grained Ti15Zr5Cu alloy was co-cultured with MC3T3-E1 cells for 1 d, 3 d and 7 d, and its cytotoxicity was evaluated using the CCK-8 Kit. As shown in Figure 8, the measured OD value increased with co-culture time, indicating osteoblast proliferation on the material's surface. Compared to the control group, the commercial pure Ti and the Ti15Zr alloy, the ultrafine-grained Ti15Zr5Cu alloy exhibited better performance in regards to promoting the proliferation of cells in the early stages (1 d, 3 d), which was because the minor release of $Cu^{2+}$ from the surface of the material contributed to the expression of an osteogenic property [29,30]. Cells that were co-cultured with the three experimental materials for 1 day had their cytoskeletons stained, as shown in Figure 9. The number of cells on the surface of the ultrafine-grained Ti15Zr5Cu alloy was significantly higher than that of the control materials, with fully extended cellular pseudopodia and good connection among cells. It can be seen that the ultrafine-grained Ti15Zr5Cu alloy is non-cytotoxic, and the minor release of $Cu^{2+}$ from the surface of the material can promote an osteogenic role in the early stage of the co-culturing test.

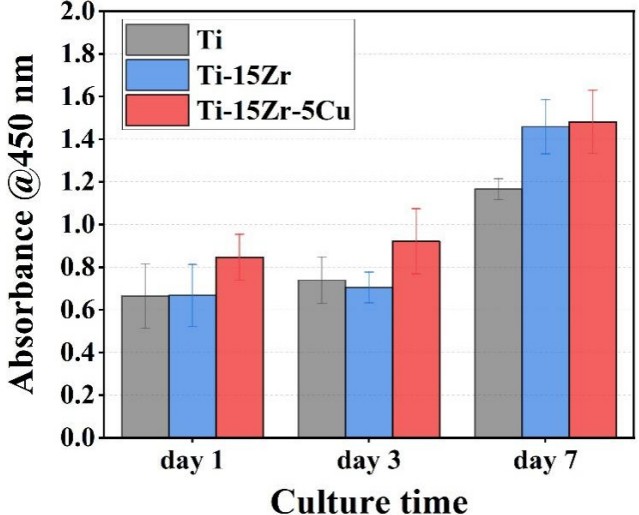

**Figure 8.** OD values of MC3T3-E1 cells co-cultured with the commercial pure Ti, the Ti15Zr alloy and the ultrafine-grained Ti15Zr5Cu alloy.

|  Ti  |  Ti-15Zr  |  Ti-15Zr-5Cu  |

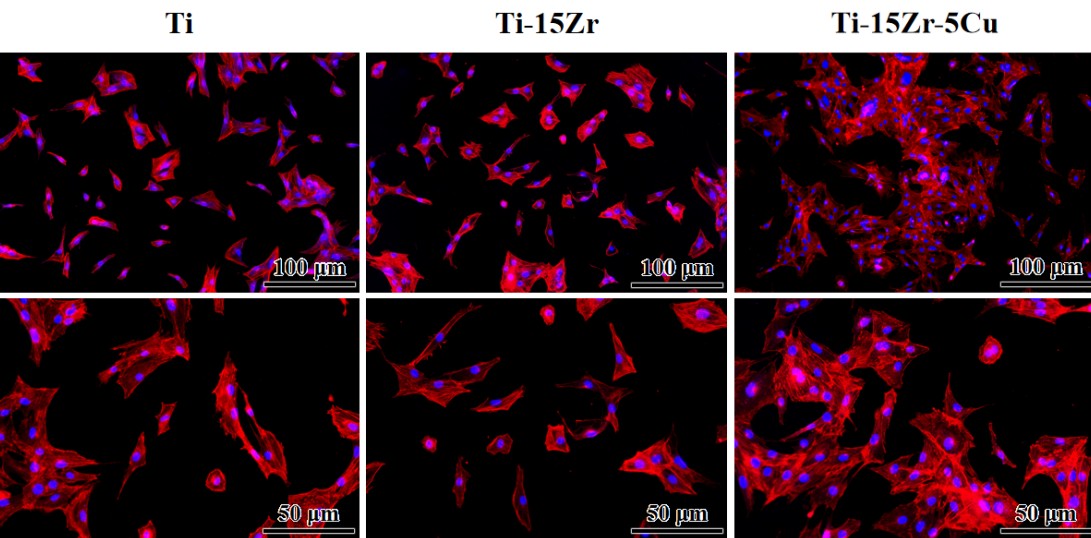

**Figure 9.** Cytoskeleton staining results of MC3T3-E1 cells co-cultured with the commercial pure Ti, the Ti15Zr alloy and the ultrafine-grained Ti15Zr5Cu alloy.

## 4. Conclusions

In this research, hot compression deformation behavior of the as-quenched Ti15Zr5Cu alloy was systematically studied, and the optimal processing window parameters of the ultrafine-grained Ti15Zr5Cu alloy were calculated. On this basis, corrosion resistance and biological properties of the prepared ultrafine-grained material were evaluated. The main conclusions are as follows:

(1) According to the flow stress–strain curves, the constitutive equation of the as-quenched Ti15Zr5Cu alloy was determined as $Z = \dot{\varepsilon} \exp(491500/RT) = 1.93 \times 10^{23} [\sinh(0.0062\sigma)]^{3.76}$. Cu and Zr alloying would increase hot deformation activation energy of $\alpha$ titanium.

(2) Processing map and SEM observation results indicated that the as-quenched Ti15Zr5Cu alloy possessed excellent hot workability and could be converted to an ultrafine-grained microstructure under the deformation condition of $0.05 \leq \dot{\varepsilon} \leq 1$ and $730\,°C \leq T \leq 750\,°C$.

(3) Electrochemical tests showed that a denser passive film formed on the surface of the ultrafine-grained Ti15Zr5Cu alloy, which contributed to better corrosion resistance than the commercial pure Ti alloy and the Ti15Zr alloy.

(4) Compared to the commercial pure Ti and the Ti15Zr alloy, the ultrafine-grained Ti15Zr5Cu alloy exhibited outstanding antibacterial properties, with an antibacterial rate against *E. coli* and *S. aureus* above 99%.

(5) Ultrafine-grained Ti15Zr5Cu alloy was non-cytotoxic and exhibited good osteogenesis in the early stage of co-culturing.

**Author Contributions:** Conceptualization, H.W. and W.G.; investigation, X.Z. and Y.L.; data curation S.Z.; writing—original draft preparation, H.W. and W.G.; supervision, K.Y. and L.R.; funding acquisition, L.R. All authors have read and agreed to the published version of the manuscript.

**Funding:** This work was financially supported by the National Key Research and Development Program of China (2018YFC1106600), Bintech-IMR R&D Program (GYY-JSBU-2022-008), Doctoral Scientific Research Foundation of LiaoNing Province (2020BS002).

**Institutional Review Board Statement:** Not applicable.

**Informed Consent Statement:** Not applicable.

**Data Availability Statement:** The data presented in this study are available on request from the corresponding author.

**Conflicts of Interest:** The authors declare no conflict of interest.

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
