# Peer review of "Enhanced Corrosion Resistance and Biological Properties of Ultrafine-Grained Ti15Zr5Cu Alloy"

_metals, doi:10.3390/met12071144_

Round 1

Reviewer 1 Report

I find the paper under review as an interesting and useful investigation for specialists in biomedical materials, but due to some small shortcomings, I suggest accepting it for publication with minor revision.

Remarks and suggestions:

Page 3, line 87: "Standard φ3 mm cylindrical tensile samples". Maybe ø instead of φ?

Page 3, lines 102-103: "It can be seen that flow stresses always increased with the increase of the deformation temperature, or with the decrease of the deformation strain rate." I see the opposite: flow stresses always decrease with increasing deformation temperature or decreasing deformation strain rate.

Page 5, line 140: It is not explained how the parameter C=1.93×10^23 was obtained in equation (11).

Page 5, Table 1: The first (left) column has a header 'Peak flow stress' but consists of values for temperature.

Page 7, lines 166-173: I would like to see references where "the strain rate sensitivity parameter m, the energy dissipation rate η, and the flow instability parameter ξ" were introduced, if they exist.

Page 7, line 173: "η reflects the ratio of the energy dissipated by the microstructure evolution during the hot deformation." The sentence is incomplete. The ratio of the dissipated energy to what energy?

Page 12, line 250: "experimental materials for 1d were stained their cytoskeletons..." What does '1d' mean?

Please pay attention to the regions of text marked in yellow in the file attached. They look like misprints, misused words, missing words, or other grammatical mistakes.

Author Response

The authors sincerely appreciate reviewers’ pertinent comments on our manuscript entitled “Fabrication of an ultrafine grained Ti15Zr5Cu alloy with good corrosion resistance and excellent biological properties” (metals-1793725). These comments are highly valuable to improve the quality of our manuscript. The authors have made a thorough revision suggested by the reviewer. Amended parts have been highlighted in the revised manuscript. The point-by-point responses to reviewers’ comments are as follows:

1. Page 3, line 87: "Standard φ3 mm cylindrical tensile samples". Maybe ø instead of φ?

Response: We have corrected it according to your comment.

2. Page 3, lines 102-103: "It can be seen that flow stresses always increased with the increase of the deformation temperature, or with the decrease of the deformation strain rate." I see the opposite: flow stresses always decrease with increasing deformation temperature or decreasing deformation strain rate.

Response: We have corrected it according to your comment.

3. Page 5, line 140: It is not explained how the parameter C=1.93×10^23 was obtained in equation (11).

Response: We have addressed this problem. In Page5, line 140, we have added "Finally, according to equation (6), the C value could be calculated by the intercept of lnÈ+Q/RT-nln[sinh(ασ)] curves."

4. Page 5, Table 1: The first (left) column has a header 'Peak flow stress' but consists of values for temperature.

Response: We have corrected the table according to your comment.

5. Page 7, lines 166-173: I would like to see references where "the strain rate sensitivity parameter m, the energy dissipation rate η, and the flow instability parameter ξ" were introduced, if they exist.

Response: We have cited references in the revised manuscript.

6. Page 7, line 173: "η reflects the ratio of the energy dissipated by the microstructure evolution during the hot deformation." The sentence is incomplete. The ratio of the dissipated energy to what energy?

Response: We have corrected the sentence. In Page 7, line 173, we have corrected it as "where η reflects the ratio of the energy dissipated by the microstructure evolution to the total input energy during the hot deformation“.

7. Page 12, line 250: "experimental materials for 1d were stained their cytoskeletons..." What does '1d' mean?

Response: We have corrected it in the revised manuscript. In page 12, line 250, we have changed "1d" to "1 day".

8. Please pay attention to the regions of text marked in yellow in the file attached. They look like misprints, misused words, missing words, or other grammatical mistakes.

Response: Thank you for your time and patient, we have corrected all the misprints, misused words, and other grammatical mistakes according to your comment.

Reviewer 2 Report

The manuscript is devoted to the description of the preparation of Ti15Zr5Cu alloy without toxic elements, which is recommended for use in the biomedical field. Its corrosion resistance, antibacterial properties and biocompatibility have been shown to be superior to those of the Ti15Zr alloy and the commercial pure Ti. The manuscript is of great theoretical and practical interest and can be published as presented.

Author Response

Thank you very much for your time and patient to read our work.

Reviewer 3 Report

The manuscript deals with the evaluation of corrosion resistance and biological properties of an ultrafine-grained Ti15Zr5Cu alloy. In this study, the authors reported the hot deformation behaviors of the material, as well as the evaluation of its corrosion resistance and biological properties.

The work presented, in general, has great value, however, there are some issues that can be improved before acceptance.

·          The proposed title "Fabrication of an ultrafine-grained Ti15Zr5Cu alloy with good corrosion resistance and excellent biological properties" is not appropriate for this article as it does not deal with the fabrication of the alloy (the fabrication was previously reported by the authors). Therefore, it would be convenient to remove the manufacturing aspect and put more emphasis on the toxicology and corrosion resistance of the sample.

·         The introduction should be rewritten to be clearer and more consistent with the research objectives. The end of the introduction seems like another abstract.

·         The methodology is not clear, it must be divided by type of test or analysis.

·         The authors must provide the bacterial strain number of Escherichia coli and Staphylococcus aureus.

·         Grammar and writing style should be improved since the vocabulary used is rather informal in the whole manuscript.

Author Response

The authors sincerely appreciate reviewers’ pertinent comments on our manuscript entitled “Fabrication of an ultrafine grained Ti15Zr5Cu alloy with good corrosion resistance and excellent biological properties” (metals-1793725). These comments are highly valuable to improve the quality of our manuscript. The authors have made a thorough revision suggested by the reviewer. Amended parts have been highlighted in the revised manuscript. The point-by-point responses to reviewers’ comments are as follows:

1. The proposed title "Fabrication of an ultrafine-grained Ti15Zr5Cu alloy with good corrosion resistance and excellent biological properties" is not appropriate for this article as it does not deal with the fabrication of the alloy (the fabrication was previously reported by the authors). Therefore, it would be convenient to remove the manufacturing aspect and put more emphasis on the toxicology and corrosion resistance of the sample.

Response:Thank you for your comment. In our previous study, we have focused on the optimum Cu content in the Ti15ZrxCu alloys,and the optimum Cu content was determined as 5wt.%. However, two important manufacturing parameters, that are, the hot deformation temperature and the deformation the strain rate were not studied in the last study. Inappropriate deformation parameters might cause the formation of coarse grains hence decreasing the mechanical properties, or resulting in the formation of holes and microcracks thus limiting the subsequent machining of the material. Therefore, it is imperative to systematically study the hot deformation behavior of the as-quenched Ti15Zr5Cu alloy and to determine the optimal hot deformation window for its manufacturing process. In this manuscript, we have paid great efforts on the manufacturing of the Ti15Zr5Cu alloy, such as the Fig. 1, Fig. 2, Fig. 3, Fig. 4, Fig. 5 and the Table 1. Therefore, it could not be suitable to delete the manufacturing part and put more emphasis on the toxicology and corrosion resistance of the sample.

2. The introduction should be rewritten to be clearer and more consistent with the research objectives. The end of the introduction seems like another abstract.

Response:Thank you for your comment. We feel sure that the introduction is consistent with the research objective of this manuscript. And at the end of the introduction should briefly introduce the purpose of the study, therefore, the present form of the introduction looks very reasonable. We feel sorry that we cannot amend the introduction section, because the other two reviewers did not suggest us to make any changes. Doing that means they have to review this manuscript again.

3. The methodology is not clear, it must be divided by type of test or analysis.

Response: According to your comment, we have added subheading in the method section in the revised manuscript. We hope this revision is acceptable.

4. The authors must provide the bacterial strain number of Escherichia coli and Staphylococcus aureus.

Response:  We have added it in the method section.

5. Grammar and writing style should be improved since the vocabulary used is rather informal in the whole manuscript.

Response: Thank you for your comment. We have improved our English basing on the comment of review 1.